# Oceanic mesoscale eddies as crucial drivers of global marine heatwaves

Ce Bian [1,2], Zhao Jing [1,2] ✉, Hong Wang [1,2], Lixin Wu [1,2], Zhaohui Chen [1,2], Bolan Gan [1,2] & Haiyuan Yang[1,2]

Marine heatwaves (MHWs) are prolonged extreme warm water events in the ocean, exerting devastating impacts on marine ecosystems. A comprehensive knowledge of physical processes controlling MHW life cycles is pivotal to improve MHW forecast capacity, yet it is still lacking. Here, we use a historical simulation from a global eddy-resolving climate model with improved representation of MHWs, and show that heat flux convergence by oceanic mesoscale eddies acts as a dominant driver of MHW life cycles over most parts of the global ocean. In particular, the mesoscale eddies make an important contribution to growth and decay of MHWs, whose characteristic spatial scale is comparable or even larger than that of mesoscale eddies. The effect of mesoscale eddies is spatially heterogeneous, becoming more dominant in the western boundary currents and their extensions, the Southern Ocean, as well as the eastern boundary upwelling systems. This study reveals the crucial role of mesoscale eddies in controlling the global MHW life cycles and highlights that using eddy-resolving ocean models is essential, albeit not necessarily fully sufficient, for accurate MHW forecasts.

Marine heatwaves (MHWs) are extreme warm water events in the ocean[1-4]. They cause severe environmental and socioeconomic impacts, including the loss of biodiversity, reduction of fishery catching rates, damage to aquaculture as well as changes in the behavior of species[5-8]. Satellite observations have revealed a significant increase in frequency, duration, and intensity of MHWs during the past several decades over most parts of the global ocean[9], primarily due to the gradual sea surface warming caused by rising greenhouse gas emissions[10]. The growing threat of MHWs on marine ecosystems underscores the imperative for a comprehensive understanding of physical mechanisms responsible for the generation, maintenance, and decay of MHWs in the global ocean[2,11-13]. Such understanding is a prerequisite for establishing a reliable forecast system for MHWs and developing sensible management strategies in a timely manner to alleviate the ecosystem stress and associated socioeconomic ramifications[14-16].

Although there are plenty of case studies focusing on the physical drivers of some major MHW events, such as the well-known Blob[17] in 2014/15 and Blob 2.0[18] in 2019 over the northeastern Pacific, global-scale analysis is still limited. Holbrook et al. [12] established the first global view of the MHW drivers based on the correlation between the observed MHW occurrence and a variety of climate modes. Furthermore, ref. 19 investigated drivers of the most extreme MHWs and found that a large fraction of MHWs in the subtropical ocean coincide with persistent atmospheric high-pressure systems and weakened surface winds. Despite the important role of air–sea interactions in the MHW life cycles implied by these observational studies, a heat budget analysis based on a global ocean-only simulation[13] reveals that the global MHWs are primarily generated by heat flux convergence of oceanic flows, whereas the sea surface heat flux is the main driver of MHW decay. However, the absence of air–sea coupling in the ocean-only simulation causes biases in the simulated sea surface temperature (SST) variability[20-22] that may further propagate into the simulated MHWs.

Oceanic mesoscale eddies with a horizontal scale from a few tens to several hundreds of kilometers, manifested in the form of fronts,

[1]Frontiers Science Center for Deep Ocean Multispheres and Earth System and Key Laboratory of Physical Oceanography, Ocean University of China, Qingdao, China. [2]Laoshan Laboratory, Qingdao, China. ✉e-mail: jingzhao@ouc.edu.cn

filaments, and coherent vortices, are the most prominent feature in the upper ocean[23,24]. They account for 70% of oceanic kinetic energy[25,26] and contribute importantly to the SST variability via their induced heat flux convergence[27–29]. Yet the effects of mesoscale eddies on the MHW life cycles in the global ocean remain unexplored and largely overlooked. In this study, we use a historical simulation from an eddy-resolving global coupled climate model (CGCM)[30] (See "CESM-H" in Methods) to evaluate the role of mesoscale eddies in the global MHW life cycles. As will be demonstrated below, the heat flux convergence by oceanic mesoscale eddies acts as a crucial driver of MHW growth and decay in the global ocean.

## Results

### Simulated MHWs in the CESM-H

Performance of the CESM-H in simulating the MHWs defined based on SST (see "Definition of MHWs" in Methods) is evaluated against satellite observations (see "Observational products" in Methods) and compared with an ensemble of state-of-the-art coarse-resolution (~1°)

CGCMs in the Coupled Model Intercomparison Project Phase 6 (CMIP6)[31] (Supplementary Table 1 and Fig. 1). The CESM-H reproduces the spatial variability of MHW intensity in the observation reasonably well (Fig. 1a, d), with the correlation coefficient between the observed and simulated spatial patterns of MHW intensity reaching 0.81. In contrast, the correlation coefficient decreases to 0.68 between the observation and CMIP6 ensemble mean, mainly due to the absence of strong MHWs in the southern hemisphere western boundary currents and their extensions (WBCEs) and the Southern Ocean in the CMIP6 CGCMs (Fig. 1g).

The CMIP6 ensemble mean underestimates the frequency but overestimates the duration of MHWs (Fig. 1k, l), which has already been noted by the existing literature[9,32]. These biases are evidently alleviated in the CESM-H, especially for the duration of MHWs. Specifically, the globally averaged MHW frequency (duration) in the CMIP6 ensemble mean biases low (high) by 8% (94%) compared to the observational counterpart, whereas this bias is reduced to 3% (59%) in the CESM-H. According to the above comparisons, we conclude that the CESM-H

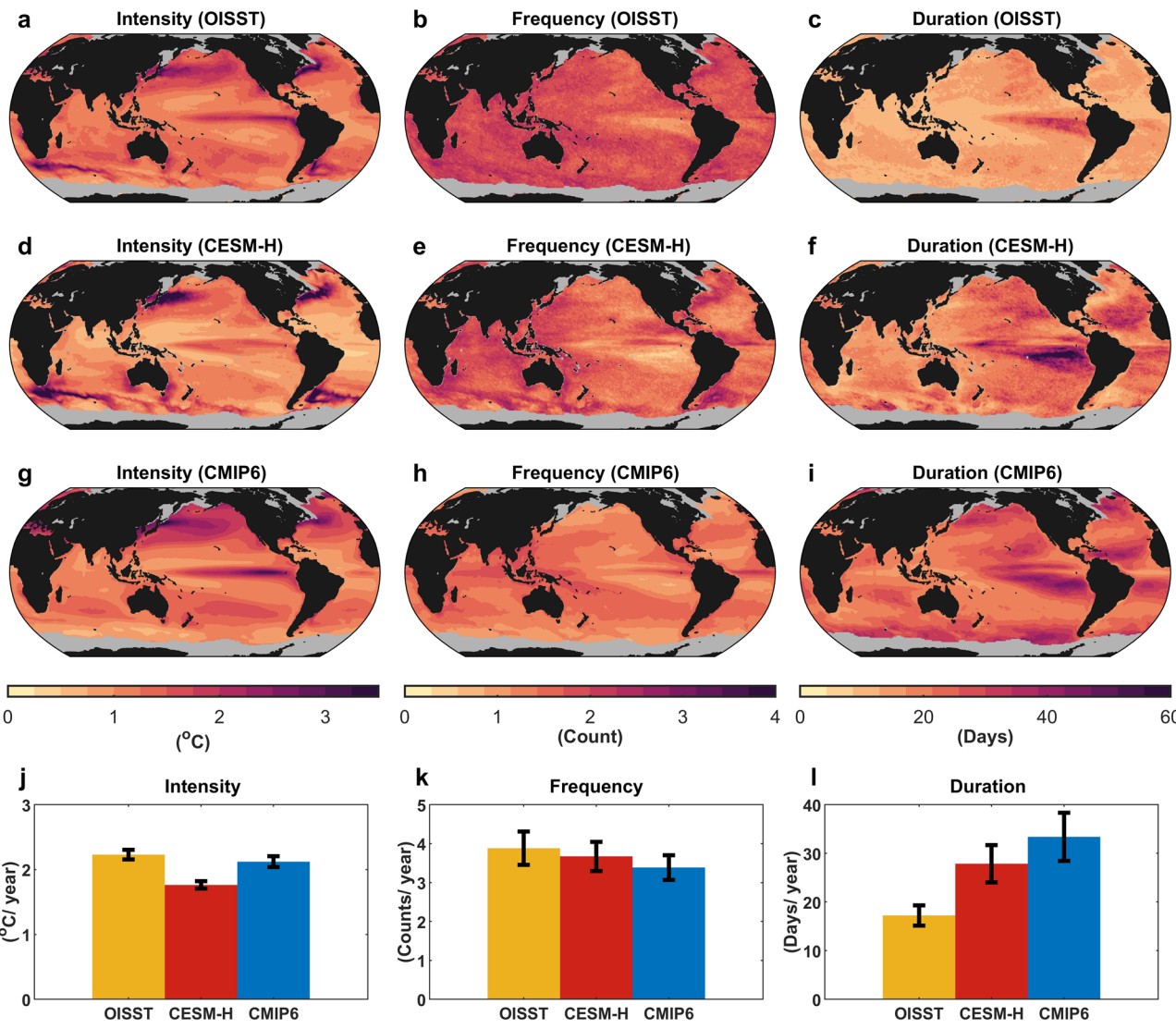

**Fig. 1 | Observed and simulated marine heatwave (MHW) statistics during 1982–2021.** Spatial distribution of climatological mean intensity (**a**), frequency (**b**), and duration (**c**) of MHWs in the observation. **d**–**f** and **g**–**i** are the same as **a**–**c**, but for the CESM-H and CMIP6 ensemble mean, respectively. MHW statistics derived from the observation and CESM-H are first smoothened using a 1° × 1° running mean and then interpolated onto the 1° × 1° regular grids, whereas MHW statistics derived from the individual CMIP6 CGCMs are interpolated onto the 1° × 1° regular grids to compute their ensemble mean. Grids with temporary or permanent sea-ice coverage in the observation are masked by gray. The bar charts in **j**–**l** show the area-weighted average of climatological mean intensity, frequency, and duration of MHWs in the global sea-ice-free ocean, respectively. The black error bars indicate the standard errors.

provides a qualitatively reliable simulation of MHWs in the global ocean, outperforming the CMIP6 ensemble mean. This lends support to the fidelity of the CESM-H in simulating the physical drivers of MHW life cycles.

## Physical divers of MHW life cycles in the global ocean

A heat budget analysis in the upper 50 m water column based on the CESM-H's diagnostic output (see "Heat budget analysis" in Methods) is used to quantify contributions to temperature changes during MHW life cycles by different physical processes, including the net sea surface heat flux (NHF), heat flux convergence by mean flows (HFC-M) and mesoscale eddies (HFC-E), and subgrid-scale mixing (MIX). To be consistent with the heat budget analysis, the MHWs are redefined based on the vertical mean temperature in the upper 50 m (denoted as $\langle T \rangle$) rather than SST (see "Definition of MHWs" in Methods). As MHWs in this study are defined based on a seasonally varying threshold[1], it is the anomaly of $\langle T \rangle$ relative to its climatological mean seasonal cycle (denoted as $\langle T_a \rangle$ henceforth) that is related to MHWs. Accordingly, all the terms in the heat budget are subtracted by their corresponding climatological mean seasonal cycles to quantify their induced changes of $\langle T_a \rangle$ during the MHW life cycles (see "Heat budget analysis" in Methods).

We perform a heat budget analysis over a fixed depth range rather than over the mixed layer, because the latter is difficult to close based on the available model output due to spatio-temporal variations of mixed layer depth (MLD). Moreover, although the heat budget analysis over the mixed layer is dynamically more suitable for analyzing the SST variability than that over a fixed depth range like 0–50 m, it is not necessarily so from biological concerns, because the depth of epipelagic zone, home to a massive number of organisms, may differ from the MLD in many parts of the global ocean[33,34].

The MHW statistics defined based on SST and $\langle T \rangle$ generally agree with each other except in the tropical eastern Pacific where the MHWs defined based on $\langle T \rangle$ are shorter and more frequent than those based on SST (Supplementary Fig. 1). Such differences may reflect the different temporal variabilities between SST and $\langle T \rangle$ due to the shallow mixed layer (shallower than 50 m) in this region but should not necessarily be interpreted as deficiencies in defining MHWs based on $\langle T \rangle$, as the epipelagic zone here is deeper than the mixed layer[33,34]. In fact, the common use of SST to define MHWs in the existing literature[1,19] is mainly due to its availability in the observation and could be insufficient to measure the thermal stress on marine ecosystems[35,36].

Figure 2a–d and e–h show the contributions of different physical processes to the $\langle T_a \rangle$ changes during the growing and decaying phases of the MHWs (Supplementary Fig. 2) averaged at each grid point, respectively. The mesoscale eddies play a crucial role in driving the MHW life cycles over most parts of the global ocean. In particular, the HFC-E accounts for 81% (74%) of $\langle T_a \rangle$ increase (decrease) during the growing (decaying) phase of the MHWs averaged over the global sea-ice-free ocean (Fig. 3a). The contribution of the HFC-E to the $\langle T_a \rangle$ change during the MHW life cycles varies in space (Fig. 2a, e). It is stronger in the WBCEs and Southern Ocean, consistent with the more energetic mesoscale eddies in these regions[23].

During the growing phase of MHWs, the HFC-M induced $\langle T_a \rangle$ increase is much smaller than that induced by the HFC-E except in the central-to-eastern equatorial Pacific, where the HFC-M plays a dominant role in driving the MHW growth (Fig. 2b). During the decaying phase of MHWs, the HFC-M still acts to increase $\langle T_a \rangle$ in many parts of the global ocean, especially in the WBCEs (Fig. 2f). The NHF causes $\langle T_a \rangle$ to decrease during the growing phase of MHWs in the WBCEs and Southern Ocean, whereas the opposite is true elsewhere (Fig. 2c). During the decaying phase, the NHF leads to a universal $\langle T_a \rangle$ decrease in the global sea-ice-free ocean (Fig. 2g). The NHF-induced $\langle T_a \rangle$ decrease reflects the sea surface heat flux feedback[37,38], i.e., the generation of sea surface heat flux anomaly by $\langle T_a \rangle$ that, in turn, damps

$\langle T_a \rangle$ itself. As to the MIX that is primarily attributed to the vertical mixing (Supplementary Fig. 3), it acts to increase $\langle T_a \rangle$ both in the growing and decaying phases of the MHWs in the WBCEs, Southern Ocean and central-to-eastern equatorial Pacific, whereas its contribution to $\langle T_a \rangle$ change is close to zero elsewhere (Fig. 2d and h).

The spatially heterogeneous effects of the different physical processes on the $\langle T_a \rangle$ changes during the MHW life cycles suggest that the dominant physical drivers of the MHW growth and decay are region-dependent (Supplementary Fig. 4). In the WBCEs with energetic mesoscale eddies, the HTC-E accounts for average for 97% (89%) of the $\langle T_a \rangle$ increase (decrease) during the growing (decaying) phase of the MHWs (Fig. 3b), acting as the single dominant driver of the MHW life cycles. Similar is the case for the Southern Ocean, with the HFC-E contributing 88% to the increase of $\langle T_a \rangle$ during the growing phase and 87% to the decrease of $\langle T_a \rangle$ during the decaying phase, respectively (Fig. 3c). In contrast, 47% of $\langle T_a \rangle$ increase during the growing phase in the central-to-eastern equatorial Pacific is attributed to the HFC-M (Fig. 3d), implying the association of MHW generation with the sea surface warming during El Niño events via the Bjerknes feedback[12,39]. Nevertheless, the HFC-E plays an important role in driving the MHW growth in this region, accounting for 45% of the $\langle T_a \rangle$ increase during the growing phase. As to the decrease of $\langle T_a \rangle$ during the decaying phase in the central-to-eastern equatorial Pacific, it is mainly ascribed to the HFC-E (70%), consistent with the damping of El Niño events by the HFC-E[40,41]. In the biologically productive eastern boundary upwelling systems[42,43], the HFC-E contributes 91% to the increase of $\langle T_a \rangle$ during the growing phase, while the decrease of $\langle T_a \rangle$ during the decaying phase is contributed primarily by the HFC-E (74%) and secondarily by the NHF (20%) (Fig. 3e). In the subtropical gyre interior, HFC-E still plays a dominant role (70%) in driving the $\langle T_a \rangle$ increase during the growing phase (Fig. 3f). During the decaying phase, the contribution to the $\langle T_a \rangle$ decrease by the NHF (44%) becomes comparable to that of the HFC-E (60%).

Finally, we re-perform the heat budget analysis by varying the lower bound of the water column from 20 m to 200 m that covers the range of euphotic zone depth over the global ocean[33,34] (Supplementary Figs. 5–7). As expected, contributions of the NHF and MIX to the $\langle T_a \rangle$ change during the MHW life cycles become more important for the shallower water column. Nevertheless, for the range of lower bound considered here, the HFC-E always plays a dominant role, lending strong support to the crucial role of oceanic mesoscale eddies in driving the global MHW life cycles.

## Role of mesoscale eddies in driving the life cycles of MHWs with large spatial scales

The HFC-E induced $\langle T_a \rangle$ change occurs primarily at the oceanic mesoscales (Supplementary Figs. 8, 9). Correspondingly, the HFC-E should be most efficient in driving the growth and decay of the MHWs whose characteristic spatial scale is comparable to that of mesoscale eddies. However, the correlation between the velocity and temperature anomalies induced by mesoscale eddies can generate HFC-E that is coherent at spatial scales larger than that of mesoscale eddies (Supplementary Figs. 8b, 9b; See "Heat budget analysis" in Methods). Such correlation can arise from the baroclinic instability[44,45], frontogenesis/frontolysis[46,47], and turbulent thermal wind[48,49]. Therefore, the HFC-E may also play a role in driving the life cycles of MHWs with a characteristic spatial scale larger than that of mesoscale eddies. To demonstrate this point, we redefine MHWs based on the large-scale $\langle T \rangle$ (denoted as $\langle \bar{T} \rangle$) that filters out the mesoscale perturbations and refer to them as the large-scale MHWs henceforth. Then we quantify the effects of different physical processes on the changes of $\langle \bar{T}_a \rangle$ during the life cycles of large-scale MHWs (See "Heat budget analysis" in Methods).

Contributions of the HFC-E to the $\langle \bar{T}_a \rangle$ changes during the growing and decaying phases of large-scale MHWs are systematically

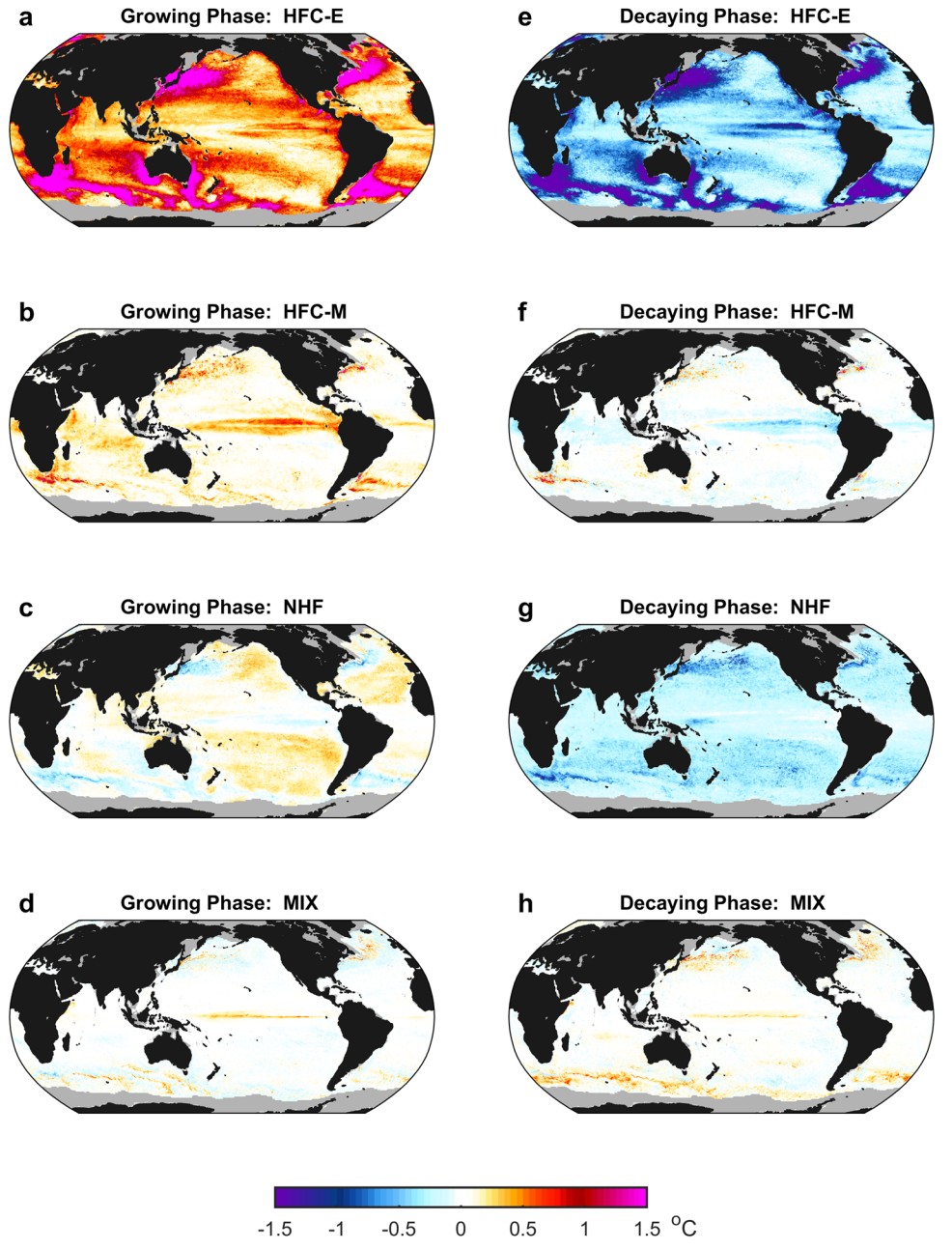

**Fig. 2 | Global distribution of the effects of different physical processes on the marine heatwave (MHW) life cycles.** Contribution to the $\langle T_a \rangle$ change during the growing phase of the MHWs averaged at each grid point by heat flux convergence of mesoscale eddies (HFC-E) (**a**) and mean flows (HFC-M) (**b**), net surface heat flux (NHF) (**c**), and subgrid-scale mixing (MIX) (**d**). **e–h** are the same as **a–d**, but for the decaying phase of the MHWs. Grids with temporary or permanent sea-ice coverage in the observation are masked by gray.

weaker than its counterparts for the all-scale MHWs (Supplementary Fig. 10), whereas the contributions of the HFC-M, NHF, and MIX are affected to a less extent by the spatially low-pass filtering (Figs. 2, 4). Correspondingly, the dominant physical drivers of the large-scale MHW life cycles in the global sea-ice-free ocean are taken over by the HFC-M and NHF for the growing phase and by the NHF alone for the decaying phase (Fig. 4). Nevertheless, the HFC-E still plays an important role in the regional large-scale MHW life cycles, contributing primarily to the $\langle \bar{T}_a \rangle$ changes during the growing and decaying phases in the WBCEs, the $\langle \bar{T}_a \rangle$ increase during the growing phase in the eastern boundary upwelling systems, and the $\langle \bar{T}_a \rangle$ decrease during the decaying phase in the central-to-eastern equatorial Pacific (Fig. 4 and Supplementary Fig. 10).

## Discussion

Our results reveal the crucial role of mesoscale eddies in driving the growth and decay of global MHWs, which is largely overlooked in the existing literature. As these mesoscale eddies are not resolved by coarse-resolution CGCMs, it may partially account for the less frequent MHWs in the CMIP6 CGCMs than the observation (Fig. 1b, h, k). In particular, the MHW frequency in the observations and CESM-H is locally increased in the WBCEs and Southern Ocean with active mesoscale eddies, whereas such increase is largely absent in the CMIP6 CGCMs (Fig. 1b, e, h). Moreover, as mesoscale eddies have a relatively shorter time scale than that of mean flows, the MHWs generated by the HFC-E are expected to last for a shorter period than those generated by the HFC-M. This may explain the, on average longer duration of MHWs

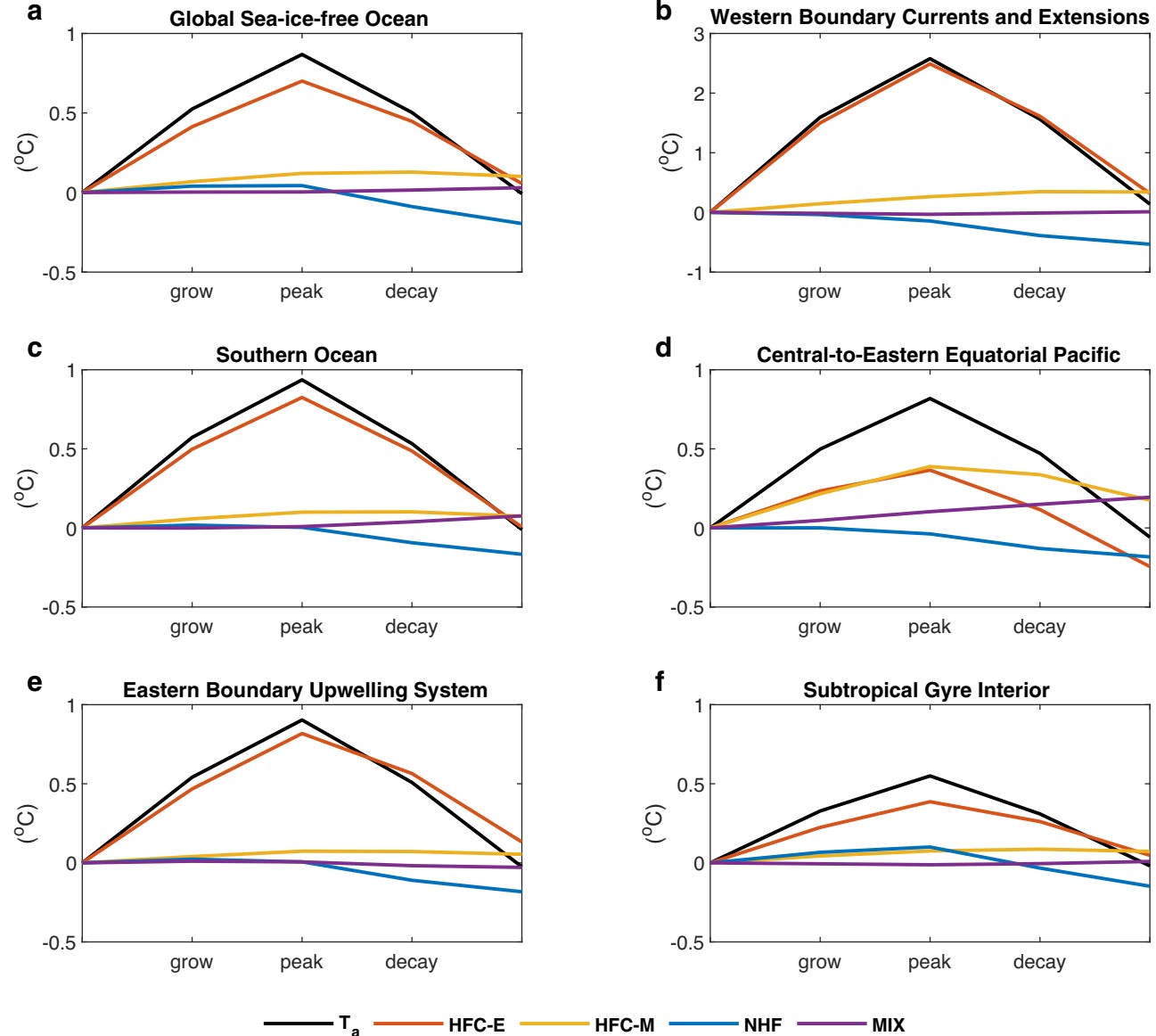

**Fig. 3 | Dominant physical drivers of the marine heatwave (MHW) life cycles in different regions.** Contribution to the $\langle T_a \rangle$ changes (black) by heat flux convergence of mesoscale eddies (HFC-E) (red) and mean flows (HFC-M) (yellow), net surface heat flux (NHF) (blue), and subgrid-scale mixing (MIX) (purple), during the growing and decaying phases of the MHWs averaged over the global sea-ice-free ocean (**a**), the western boundary currents and their extensions (**b**), the Southern Ocean (**c**), the central-to-eastern equatorial Pacific (**d**), the eastern boundary upwelling systems (**e**), and the subtropical gyre interior (**f**), respectively. Domains of the different regions are marked in Supplementary Fig. 4.

in the CMIP6 CGCMs than the CESM-H and observations (Fig. 1c, f, i, l). Nevertheless, we remark that the overly small MHW number and overly long MHW duration still persist in the CESM-H albeit alleviated. Such biases in the CESM-H may partially result from the unresolved heat flux convergence by submesoscale eddies[50] and deficiencies of vertical mixing parameterization[51] that do not account for the effects of surface waves[52,53] and Langmuir turbulence[54].

The mesoscale eddies, also known as the geostrophic turbulence[55], are essentially chaotic with limited predictability. This imposes a strong restriction on the forecast capacity of MHWs driven by mesoscale eddies and is consistent with the recent finding[16] that the forecast skills of MHWs by numerical models are evidently lower in the eddy-rich regions than elsewhere[12,16,56]. For these mesoscale eddy-driven MHWs, it is more feasible to predict their statistics instead of individual characteristics. It has been well recognized that mesoscale eddies exhibit evident variabilities on multiple time scales regulated

by changes in large-scale oceanic and atmospheric circulations[57–60]. How the variabilities of mesoscale eddies may affect the statistics of MHWs, remain unexplored but will be pivotal for proactive decision-making.

## Methods
### CESM-H
This study uses a global climate simulation based on the Community Earth System Model (CESM) with a 0.1° horizontal resolution for the ocean and sea-ice components and a 0.25° horizontal resolution for the atmosphere and land components. The CESM-H has a 250-year historical and future transient climate (HF-TNST) experiment for 1850–2100 following the design protocol of the Coupled Model Intercomparing Project phase 5 (CMIP5) experiments[61]. Specially, the simulation is branched off from the 250th year of the pre-industrial control simulation and forced by the historical forcing from 1850 to

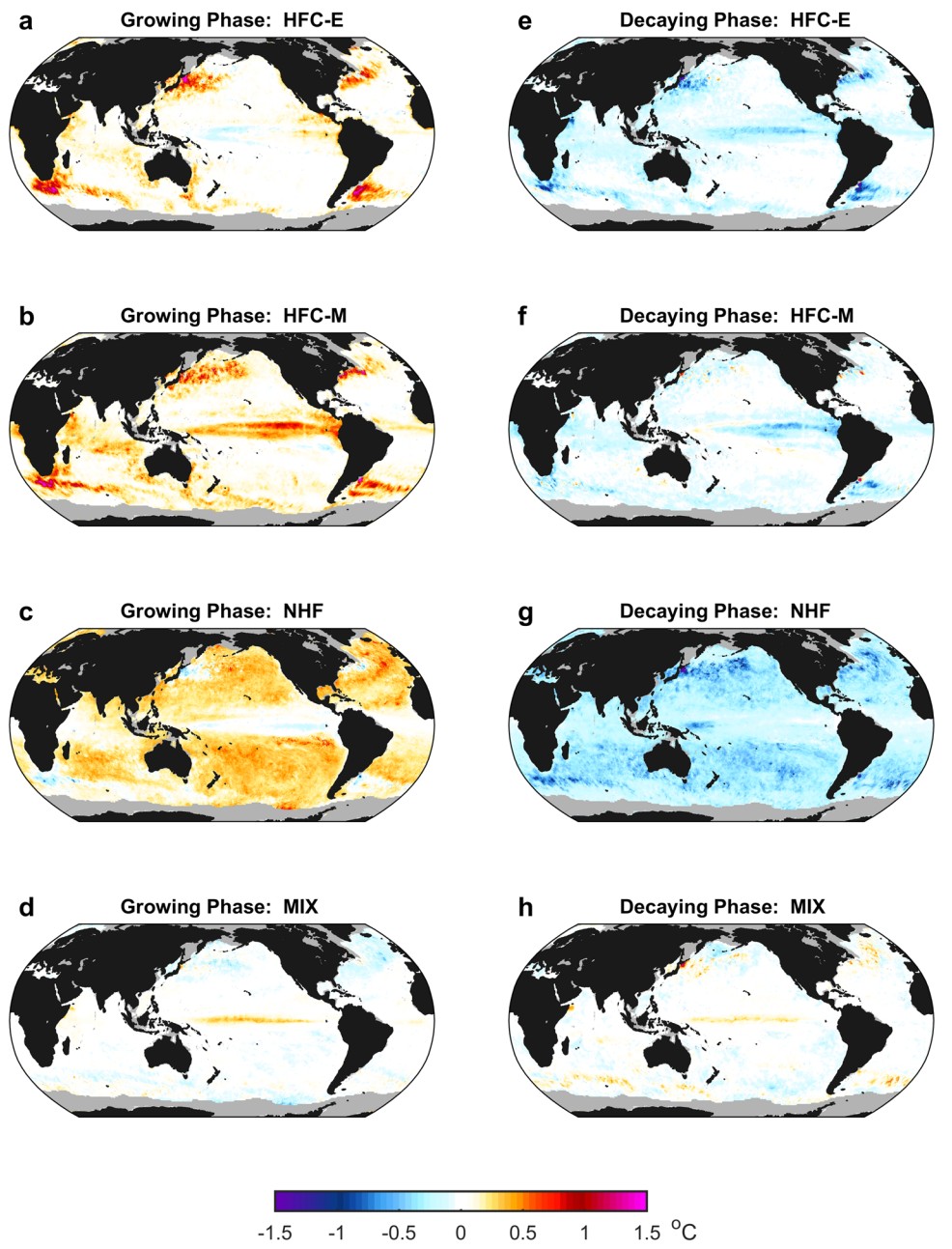

**Fig. 4 | Global distribution of the effects of different physical processes on the life cycles of marine heatwaves (MHWs) with a spatial scale larger than that of mesoscale eddies.** Contribution to the $\langle T_a \rangle$ change during the growing phase of the large-scale MHWs averaged at each grid point by heat flux convergence of mesoscale eddies (HFC-E) (**a**) and mean flows (HFC-M) (**b**), net surface heat flux (NHF) (**c**), and subgrid-scale mixing (MIX) (**d**). **e**–**h** are the same as **a**–**d**, but for the decaying phase of the MHWs. The large-scale MHWs are defined based on the $3° \times 3°$ horizontal running mean $\langle T \rangle$ and the individual terms in the heat budget are low-pass filtered using a $3° \times 3°$ horizontal running mean. Grids with temporary or permanent sea-ice coverage in the observation are masked by gray.

2005, followed by concentration pathway 8.5 (RCP8.5) forcing during 2006–2100[30]. The CESM-H saves the daily SST during 1877–2100. In addition, there is a complete daily diagnostic output of temperature governing equation during 1920–1934.

### Observational products
The observational SST comes from the National Oceanic and Atmospheric Administration (NOAA) Optimum Interaction Sea Surface Temperature V2.0 high-resolution (OISST), which is derived from the advanced very high-resolution radiometer (AVHRR). The daily SST is provided on a 0.25° × 0.25° spatial grid. The data from Jan 1982–Dec 2021 are used for analysis.

### Definition of MHWs
An MHW is defined as an event with at least five contiguous days of a given temperature index above its seasonally varying 90th percentile calculated over a baseline period[1]. The temperature index is chosen as SST and the baseline period is set as 1982–2021 for the comparison of the MHWs between the observation and CGCM simulations. However, the temperature index is chosen as the vertical mean temperature in the upper 50 m and the baseline period is set as 1920–1934 when analyzing the physical processes governing the MHW life cycles, as there is no complete daily diagnostic output of temperature governing equation during 1982–2021. The frequency, duration, and mean intensity of an MHW are computed following ref. 1.

The geographic distributions of MHW statistics computed by using different temperature indices and baseline periods are qualitatively consistent with each other (Supplementary Fig. 1). As greenhouse warming is insignificant before the 1950s, such consistency implies that by now, the anthropogenic climate changes have not altered the physical processes underpinning the MHWs substantially, lending supports that the results derived for the period 1920–1934 are also representative of the present-day situation.

The start and end times of an MHW are typically defined as the time when the temperature index rises above and declines below its threshold (denoted as $t_{s0}$ and $t_{e0}$ hereinafter), respectively[1] (Supplementary Fig. 2). However, we remark that such defined start and end times may not be suitable for analyzing the physical drivers of MHW life cycles. For instance, the temperature index may have already increased by an evident amount before it exceeds the threshold and it is the physical process driving this increase that is mainly responsible for the MHW generation. Therefore, we define the start time $t_s$ of an MHW as the local minimum point just before $t_{s0}$ and its end time $t_e$ as the local minimum point just after $t_{e0}$ (Supplementary Fig. 2). The peaking time $t_p$ of an MHW is defined as the maximum point of the temperature index anomaly relative to its climatological mean seasonal cycle. The period between the start and peak times of an MHW is defined as its growing phase, while the period between the peak and end times is defined as its decaying phase.

### Heat budget analysis

To evaluate the physical drivers of MHW life cycles, a heat budget in the upper 50 m is performed:

$$\left\langle \frac{\partial T}{\partial t} \right\rangle = \left\langle -\nabla \cdot (\bar{\mathbf{u}}\bar{T}) \right\rangle + \left\langle -\nabla \cdot (\mathbf{u}'T') - \nabla \cdot (\bar{\mathbf{u}}T') - \nabla \cdot (\mathbf{u}'\bar{T}) \right\rangle \\ + \frac{Q_{\mathrm{NHF}}}{\rho_0 C_p H} + \langle \mathrm{HMIX} \rangle + \langle \mathrm{VMIX} \rangle \tag{1}$$

where $T$ is the temperature, $\mathbf{u} = (u,v,w)$ is the three-dimensional velocity, $\nabla$ is the three-dimensional gradient operator, $Q_{\mathrm{NHF}}$ is the surface heat flux into the ocean minus the solar radiation penetrated out of the layer base at 50 m, $\rho_0 = 1026\,\mathrm{kg \cdot m^{-3}}$ is the reference seawater density, $C_p = 4000\,\mathrm{J(kg \cdot K)^{-1}}$ is the heat capacity of seawater, $H = 50\,\mathrm{m}$ is the layer thickness, HMIX is the subgrid-scale horizontal mixing, VMIX is the subgrid-scale vertical mixing parameterized by a $K$-profile parameterization[51] the overbar denotes the mean flow signals obtained by a $3° \times 3°$ horizontal running mean, the prime denotes the mesoscale eddy field computed as the perturbation from the $3° \times 3°$ horizontal running mean, and the angle brackets denote the vertical average in the upper 50-m layer. The term on the left-hand side of Eq. (1) is the temperature tendency. The first $\left\langle -\nabla \cdot (\bar{\mathbf{u}}\bar{T}) \right\rangle$ and second terms $\left\langle -\nabla \cdot (\mathbf{u}'T') - \nabla \cdot (\bar{\mathbf{u}}T') - \nabla \cdot (\mathbf{u}'\bar{T}) \right\rangle$ on the right-hand side are the heat flux convergence by mean flows and mesoscale eddies (HFC-M and HFC-E), respectively. The third term is the temperature change caused by the net sea surface heat flux into the upper 50-m water column (NHF). All the terms in Eq. (1) can be explicitly computed based on the CESM-H's diagnostic output. As it is the anomaly of $\langle T \rangle$ (denoted as $\langle T_a \rangle$) relative to its climatological mean seasonal cycle that is related to MHWs by definition[1], the climatological mean seasonal cycles are subtracted from all the terms in Eq. (1). Then individual terms in Eq. (1) are integrated over the growing or decaying phase of MHWs to quantify the contributions of different physical processes to the changes of $\langle T_a \rangle$. To analyze the physical drivers of MHW life cycles at a given grid point (Fig. 2), change of $\langle T_a \rangle$ and its decomposition into components contributed by different processes are averaged over all the MHWs at that grid point. For the physical drivers of MHW life cycles in a given region (Fig. 3), change of $\langle T_a \rangle$ and its contributions

by different processes are weighted average over all the MHWs at the grid points within that region, with the grid area taken as the weight.

It should be noted that mesoscale eddies can cause temperature change with a spatial scale larger than that of mesoscale eddies via the correlation between $\mathbf{u}'$ and $T'$. This can be shown by horizontally averaging Eq. (1) using the $3° \times 3°$ horizontal running mean:

$$\left\langle \overline{\frac{\partial T}{\partial t}} \right\rangle = \overline{\left\langle -\nabla \cdot (\bar{\mathbf{u}}\bar{T}) \right\rangle} + \overline{\left\langle -\nabla \cdot (\mathbf{u}'T') - \nabla \cdot (\bar{\mathbf{u}}T') - \nabla \cdot (\mathbf{u}'\bar{T}) \right\rangle} \\ + \overline{\frac{Q_{\mathrm{NHF}}}{\rho_0 C_p H}} + \langle \overline{\mathrm{HMIX}} \rangle + \langle \overline{\mathrm{VMIX}} \rangle \tag{2}$$

The term $-\nabla \cdot (\bar{\mathbf{u}}T') - \nabla \cdot (\mathbf{u}'\bar{T})$ is largely suppressed by the $3° \times 3°$ horizontal running mean, whereas the term $-\nabla \cdot (\mathbf{u}'T')$ is less so due to the correlation between $\mathbf{u}'$ and $T'$.

## Data availability

Source data are provided with this paper. The OISSTv2 data are provided by NOAA from their website https://psl.noaa.gov/data/gridded/data.noaa.oisst.v2.highres.html. The CMIP6 CGCM model data can be downloaded from https://esgfnode.llnl.gov/projects/cmip6/. The CESM-H data used in this work are available from https://ihesp.github.io/archive/products/ds_archive/Sunway_Runs.html.

## Code availability

The iHESP version of CESM-H code is available at ZENODO via https://doi.org/10.5281/zenodo.3637771. The code used to analyze these data and generate the results presented in the study can be obtained from https://github.com/cecbian/MHW. The Matlab2022b is used for plotting. The MATLAB code of MHW distinguish is obtained from https://github.com/ZijieZhaoMMHW/m_mhw1.0[62].

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

## Acknowledgements

This work was supported by the Science and Technology Innovation Foundation of Laoshan Laboratory (No. LSKJ202202501 to Z.J.) and Taishan Scholar Funds (tsqn201909052 to Z.J.). Computational resources were supported by Laoshan Laboratory.

## Author contributions

C.B. conducted the analysis under Z.J.'s instruction. Z.J. wrote the manuscript and proposed the central idea. H.W. performed the CESM-H simulation. L.W. led the research and organized the writing of the manuscript. Z.C., B.G., and H.Y. were involved in interpreting the results and contributed to improving the manuscript.

## Competing interests

The authors declare no competing interests.
