## [Peer Review File · Nature Communications]

Oceanic Mesoscale Eddies as Crucial Drivers of Global Marine HeatwavesREVIEWER COMMENTS

Reviewer #1 (Remarks to the Author):

The study of Bian et al. presents analyses of high-resolution simulations to evaluate the drivers of Marine Heatwaves. The authors have done a set of analyses and presented an interpretation that will be of broad interest to the climate community, However, I do believe that in order for the study to achieve the standard required for acceptance at this journal, the work should be more clearly described in terms of its relation to ocean observations and to published studies on alternative mechanisms. The authors also need to make sure that they make fully appropriate use of data constraints, including the MLD constraint suggested below.

Major Comments:

First and foremost, I think that it is very important for the authors to clarify that although they have emphasized geostrophic turbulence (mesoscale eddies) here, another type of turbulence in the form of shear-turbulence can have important impacts on mixed layer properties, and should not be ignored. The studies of Qiao et al., GRL, 2004 ; Huang and Qiao, JGR; 2010 ; and Huang et al., JGR, 2012 have pointed to this kind of shear-turbulence process being of first importance for determining mixed layer properties. Am I correct in assuming that the CESM-HR model does not include a wave model or another appropriate parameterization to account for this? This really needs to be stated. I don't think that this should in any way interfere in itself with the publishing of this very interesting study at hand, but this context is important.

As for the heat budget analysis here, I think that this is of great value and interest, but I also think that it is important to include some clarification to ensure that the manuscript is truly accessible. First, the authors have chosen to perform their heat budget analysis over a depth range of 0m-50m. This is clearly then not a clean analysis over mixed layer, which I can understand would be somewhat prohibitive for a model with this kind of resolution. But it should be stated explicitly why a full mixed layer analysis is not presented, with this statement in the main text. I think that again in terms of clarity of presentation, the authors will also need to show how their mixed layer depth compares to observational constraints. Depending on the frequency of output of either HBLT (for the POP model) or a density-threshold MLD, it would be very helpful to show two things: (i) how does the climatological seasonal minimum in MLD compare with observations in a map presentation, and (ii) for how many days per year are MLDs less than 50m in the simulations? And how does this impact the interpretation?

The key point that links the above comments is that with storms etc. one finds that intermittent events (entrainment etc.) should be expected to be a missing process in CESM1. Perhaps entrainment as destratification (cooling) would preferentially impact the duration of MHWs, through the ability to provide rapid cooling. I think that with a couple of figures (suggested above) and broader contextualizing would serve to strengthen the already strong and impressive message presented in the manuscript.

Minor Comments:

Line 29 (within abstract): "Using a global..." should be "Using historical simulations with a global..."

Line 34: "spatial scale" should say "characteristic spatial scale"

Lines 37-40: I think that the final sentence needs to be more mindful of the Major Points above. One way of stating this would be:

"This study reveals a first-order role of mesoscale eddies in controlling MHW life cycles, and highlights that using eddy-resolving ocean numerical models should be a necessary, though not necessarily fully sufficient, condition for accurate MHW forecasts"

Line 58: I would suggested changing "imposes imperative demand" to "underscores the

imperative"

Line 61: I would change "of MHWs and making" to "for MHWs and developing"

Line 67: establish => established

Line 69: investigate => investigated

Line 73: "ocean-alone" => "ocean-only"

Lines 84-85: the text really needs to state the type of historical run that has been conducted, with appropriate references.

Line 90: validated => evaluated

]Finally, to reiterate, I wish to emphasize that the authors are to be commended for a very strong study that should be of broad interest, and the above comments are intended to recommend Minor revisions. The methodologies are sound, the paper is well presented, there aren't flaws in the data analysis, and there is sufficient description for the results to be verified with access to the model output.

Reviewer #2 (Remarks to the Author):

The manuscript is concerned with origins of Marine Heat Waves (MHWs) and is based on coupled simulations with high resolution in the ocean (CESM-H). The authors first demonstrate improved fidelity of MHW simulations in the model, and then analyze the heat budget in the upper 50 meters of the ocean, with a particular focus on the importance of mesoscale eddies. The results demonstrate a leading role of eddy-induced heat advection in a life cycle of MHWs in several regions, including western boundary regions, the Southern Ocean and eastern boundary regions.

Origins of MHWs is one of the "hot" topics in climate science, and this paper will be relevant to a large group of scientists interested in ocean dynamics, climate variability and extreme events. The results are convincing, and the presentation is concise and clear. I have several specific questions on methodology and conclusions, that will need to be addressed before this manuscript can be published.

Questions:

Why is the baseline period for the studies of physical processes chosen as 1920-1934 and not a more recent period? Is it because of data availability or other reasons?

Does HFC-E include mixed eddy-mean terms, that is, 3rd and 4th terms on the right-hand side of Eq. (1)?

Why does MIX act to increase T_a ? I expect vertical mixing to be always cooling, since it represents a diffusive exchange with the colder ocean below 50 m.

l.195: I do not understand the difference between Figs. 2 and 4. The analysis of large-scale MHWs is done by low pass filtering all terms in the Eq. (1), which should simply smooth the HFC-E. The smoothing cannot dramatically decrease the amplitude of HFC-E, unless the high-resolution field is extremely noisy and/or has alternating signs; none of these properties are visible in, for example, the Kuroshio region in Fig.2. It is possible that I am missing something, but the text needs to explain the difference.

Minor issues:

l.34: "... is comparable or larger than the scale of eddies"

l. 37: comma before "as well as"

l.209 "the" is not needed

l.211: "in observations"

l.213-217: This sentence should be shortened or broken in two sentences.

l.294: it is better to say "climatological mean is subtracted from all the terms in Eq. (1)"

l.309: a prime is missing next to "u" in the formula

Reply to the first reviewer

We are very grateful to you for your time in carefully reading our manuscript and providing helpful comments that make our manuscript better. We have carefully considered each of your comments (in blue) and revised the manuscript accordingly. Please find our response (in black) to your comments below.

Reviewer #1 (Remarks to the Author):

The study of Bian et al. presents analyses of high-resolution simulations to evaluate the drivers of Marine Heatwaves. The authors have done a set of analyses and presented an interpretation that will be of broad interest to the climate community, However, I do believe that in order for the study to achieve the standard required for acceptance at this journal, the work should be more clearly described in terms of its relation to ocean observations and to published studies on alternative mechanisms. The authors also need to make sure that they make fully appropriate use of data constraints, including the MLD constraint suggested below.

Major Comments:

1. First and foremost, I think that it is very important for the authors to clarify that although they have emphasized geostrophic turbulence (mesoscale eddies) here, another type of turbulence in the form of shear-turbulence can have important impacts on mixed layer properties, and should not be ignored. The studies of Qiao et al., GRL, 2004 ; Huang and Qiao, JGR; 2010 ; and Huang et al., JGR, 2012 have pointed to this kind of shear-turbulence process being of first importance for determining mixed layer properties. Am I correct in assuming that the CESM-HR model does not include a wave model or another appropriate parameterization to account for this? This really needs to be stated. I don't think that this should in any way interfere in itself with the publishing of this very interesting study at hand, but this context is important.

We are grateful to you for pointing out this important issue. Like many other CGCMs, the CESM-HR does not parameterize the surface wave-induced mixing, which may cause some uncertainties in the simulated SST and MLD variabilities and partially account for the MHW bias

in the CESM-HR. We have clearly stated this limitation and cited the references you mentioned in the revised manuscript. Please see Line 233-236.

Following your comment, we have compared the summer and winter MLD in the observation, CESM-H and ensemble mean of CMIP6 CGCMs (Fig. R1). Climatological mean MLD in different seasons simulated by the CESM-H is qualitatively consistent with its observational counterpart despite some quantitative differences. The root mean square error (RMSE) of MLD in the CESM-H is comparable to that in the ensemble mean of CMIP6 CGCMs and thus at an acceptable level. Moreover, the MLD in the CSEM-H is generally larger than the observed value especially in the Southern Ocean in austral winter (Fig. R2), implying that the parameterized vertical mixing in the CESM is unlikely to be severely underestimated. Furthermore, as will be demonstrated in our reply to your next comment, the uncertainties in the simulated MLD do not degrade the leading-order role of oceanic mesoscale eddies in driving the global MHW life cycles.

Fig. R1 Climatological mean MLD in boreal summer (JJA) and winter (DJF) derived from the Argo observations (a,d), CESM-H (b,e) and ensemble mean of CMIP6 CGCMs (c,f). The MLD is defined as the depth where the potential density is 0.03 kg/m^3 larger than the surface value. The numbers in the brackets are the root mean square error of MLD in the numerical model.

Fig. R2 Climatological mean MLD in boreal summer (JJA) and winter (DJF) derived from the CESM-H (a,c) and ensemble mean of CMIP6 CGCMs (b,d) minus their counterparts from the Argo observations. The MLD is defined as the depth where the potential density is 0.03 kg/m^3 larger than the surface value.

2. As for the heat budget analysis here, I think that this is of great value and interest, but I also think that it is important to include some clarification to ensure that the manuscript is truly accessible. First, the authors have chosen to perform their heat budget analysis over a depth range of 0m-50m. This is clearly then not a clean analysis over mixed layer, which I can understand would be somewhat prohibitive for a model with this kind of resolution. But it should be stated explicitly why a full mixed layer analysis is not presented, with this statement in the main text. I think that again in terms of clarity of presentation, the authors will also need to show how their mixed layer depth compares to observational constraints. Depending on the

frequency of output of either HBLT (for the POP model) or a density-threshold MLD, it would be very helpful to show two things: (i) how does the climatological seasonal minimum in MLD compare with observations in a map presentation, and (ii) for many days per year are MLDs less than 50m in the simulations? And how does this impact the interpretation?

Thanks for your comment. The spatio-temporal variations of MLD make it difficult to close the heat budget over the mixed layer based on the available model output. Moreover, although the heat budget analysis over the mixed layer is dynamically more suitable than that over a fixed depth range like 0-50 m, it is not necessarily so from biological concerns, because the depth of euphotic or epipelagic zone, home to a massive number of organisms, may differ evidently from the MLD in many parts of the global ocean (Siegel et al. 2014; Levinton 2014).

We have to admit that using a fixed depth of 50 m for the heat budget analysis is somewhat arbitrary. Considering that the research interest in the MHWs is largely rooted from their biological effects, we have performed sensitivity analysis in the revised manuscript by varying the depth from 20 m to 200 m covering the range of euphotic zone depth over the global ocean (Siegel et al. 2014; Levinton 2014). Figs. R3-6 compare the contribution of different processes to the MHW growth and decay over the upper 20-m, 50-m, 100-m and 200-m water columns, respectively. As expected, the effects of vertical mixing and sea surface heat flux become more important for shallower water columns. Nevertheless, for all the depths considered here, heat flux convergence by oceanic mesoscale eddies always plays a dominant role. This lends strong support to the major conclusions of our manuscript.

In the revised manuscript, we have described in the main text the reasons why a mixed-layer heat budget analysis is not done (See line 122-129). We also added the sensitivity analysis to the revised manuscript. Please see Line 185-191 and Supplementary Figs. 5-7.

Fig. R3 Contribution to the $\langle T_a \rangle$ changes by different physical processes during the growing and decaying phases of the MHWs averaged over the global sea-ice-free ocean (a), the western boundary currents and their extensions (b), the Southern Ocean (c), the central-to-eastern equatorial Pacific (d), the eastern boundary upwelling systems (e), and the subtropical gyre interior (f), respectively. Here $\langle \rangle$ represents the vertical average over the upper 20 m.

Fig. R4 Same as Fig. R3 but $\langle \rangle$ represents the vertical average over the upper 50 m.

Fig. R5 Same as Fig. R3 but $\langle \rangle$ represents the vertical average over the upper 100 m.

Fig. R6 Same as Fig. R3 but $\langle \rangle$ represents the vertical average over the upper 200 m.

3. The key point that links the above comments is that with storms etc. one finds that intermittent events (entrainment etc.) should be expected to be a missing process in CESM1. Perhaps entrainment as destratification (cooling) would preferentially impact the duration of MHWs, through the ability to provide rapid cooling. I think that with a couple of figures (suggested above) and broader contextualizing would serve to strengthen the already strong and impressive message presented in the manuscript.

Thanks for your advice. It is a common problem that CGCMs including the CESM-HR simulate overly-long MHWs compared to the observations (Fig. 11). The entrainment processes are parameterized by the K-profile scheme (Large et al. 1994) in the CESM-HR. However, as the KPP

scheme does not parameterize the effect of Langmuir turbulence, it is likely to be insufficient for representing the entrainment processes (Grant and Belcher 2009; McWilliams et al. 2014). This may partially account for the bias of MHW duration. We have briefly described this issue in the revised manuscript. Please see Line 233-236.

Minor Comments:

Line 29 (within abstract): “Using a global...” should be “Using historical simulations with a global....”

Revised.

Line 34: “spatial scale” should say “characteristic spatial scale”

Revised.

Lines 37-40: I think that the final sentence needs to be more mindful of the Major Points above.

One way of stating this would be:

“This study reveals a first-order role of mesoscale eddies in controlling MHW life cycles, and highlights that using eddy-resolving ocean numerical models should be a necessary, though not necessarily fully sufficient, condition for accurate MHW forecasts”

Thanks for your advice. This sentence has been revised as:

“This study reveals the crucial role of mesoscale eddies in controlling the global MHW life cycles and highlights that using eddy-resolving ocean models should be essential, albeit not necessarily fully sufficient, for accurate MHW forecasts.”

Line 58: I would suggested changing “imposes imperative demand” to “underscores the imperative”

Revised.

Line 61: I would change “of MHWs and making” to “for MHWs and developing”

Revised.

Line 67: establish => established

Revised.

Line 69: investigate => investigated

Revised.

Line 73: “ocean-alone” => “ocean-only”

Revised.

Lines 84-85: the text really needs to state the type of historical run that has been conducted, with appropriate references.

The historical simulation covers the period 1850-2005 and follows the design protocol of the Coupled Model Intercomparing Project phase 5 (CMIP5) experiments (Taylor et al. 2012). We have added the above information in the method section. Please see Line 251-256.

Line 90: validated => evaluate

Revised.

Finally, to reiterate, I wish to emphasize that the authors are to be commended for a very strong study that should be of broad interest, and the above comments are intended to recommend Minor revisions. The methodologies are sound, the paper is well presented, there aren't flaws in the data analysis, and there is sufficient description for the results to be verified with access to the model output.

Thanks for your highly positive comments.

Reference list

- Grant, A. L. M., and S. E. Belcher, 2009: Characteristics of Langmuir Turbulence in the Ocean Mixed Layer. *Journal of Physical Oceanography*, **39**, 1871–1887, <https://doi.org/10.1175/2009JPO4119.1>.
- Large, W. G., J. C. McWilliams, and S. C. Doney, 1994: Oceanic vertical mixing: A review and a model with a nonlocal boundary layer parameterization. *Rev. Geophys.*, **32**, 363, <https://doi.org/10.1029/94RG01872>.
- Levinton, J. S., 2014: *Marine biology: function, biodiversity, ecology*. Fourth edition. Oxford University Press, 516 pp.

- McWilliams, J. C., E. Huckle, J. Liang, and P. P. Sullivan, 2014: Langmuir Turbulence in Swell. *Journal of Physical Oceanography*, **44**, 870–890, <https://doi.org/10.1175/JPO-D-13-0122.1>.
- Siegel, D. A., K. O. Buesseler, S. C. Doney, S. F. Sailley, M. J. Behrenfeld, and P. W. Boyd, 2014: Global assessment of ocean carbon export by combining satellite observations and food-web models. *Global Biogeochem. Cycles*, **28**, 181–196, <https://doi.org/10.1002/2013GB004743>.
- Taylor, K. E., R. J. Stouffer, and G. A. Meehl, 2012: An Overview of CMIP5 and the Experiment Design. *Bulletin of the American Meteorological Society*, **93**, 485–498, <https://doi.org/10.1175/BAMS-D-11-00094.1>.

Reply to the second reviewer

We are very grateful to you for your time in carefully reading our manuscript and providing helpful comments that make our manuscript better. We have carefully considered each of your comments (in blue) and revised the manuscript accordingly. Please find our response (in black) to your comments below.

Reviewer #2 (Remarks to the Author):

The manuscript is concerned with origins of Marine Heat Waves (MHWs) and is based on coupled simulations with high resolution in the ocean (CESM-H). The authors first demonstrate improved fidelity of MHW simulations in the model, and then analyze the heat budget in the upper 50 meters of the ocean, with a particular focus on the importance of mesoscale eddies. The results demonstrate a leading role of eddy-induced heat advection in a life cycle of MHWs in several regions, including western boundary regions, the Southern Ocean and eastern boundary regions.

Origins of MHWs is one of the “hot” topics in climate science, and this paper will be relevant to a large group of scientists interested in ocean dynamics, climate variability and extreme events. The results are convincing, and the presentation is concise and clear. I have several specific questions on methodology and conclusions, that will need to be addressed before this manuscript can be published.

Questions:

1. Why is the baseline period for the studies of physical processes chosen as 1920-1934 and not a more recent period? Is it because of data availability or other reasons?

We choose 1920-1934 period for budget analysis simply because of the data availability. When running the high-resolution CESM, we saved complete daily diagnostic output of temperature governing equation only for a limited period due to the large storage burden. We have made this clear in the revised manuscript. Please see Line 268-271.

We compare the MHW statistics computed using the CESM-simulated SST during 1982-2021 and vertical mean temperature in the upper 50 m during 1920-1934. The geographic distributions of MHW statistics computed by using different temperature indices and baseline

periods are qualitatively consistent with each other (Supplementary Fig. 1). As the greenhouse warming is insignificant before the 1950s, such consistency implies that by now the anthropogenic climate changes have not altered the physical processes underpinning the MHWs substantially, lending supports that the results derived for the period 1920-1934 are also representative of the present-day situation.

2. Does HFC-E include mixed eddy-mean terms, that is, 3rd and 4th terms on the right-hand side of Eq. (1)?

Yes, HFC-E in this manuscript is defined as $\langle -\nabla \cdot (\mathbf{u}'T') - \nabla \cdot (\bar{\mathbf{u}}T') - \nabla \cdot (\mathbf{u}'\bar{T}) \rangle$ which includes the mixed eddy-mean terms (Please see Line 307-309). We split the HFC-E into $\langle -\nabla \cdot (\mathbf{u}'T') \rangle$ and $\langle -\nabla \cdot (\bar{\mathbf{u}}T') - \nabla \cdot (\mathbf{u}'\bar{T}) \rangle$, and find that the former makes dominant contribution to the HFC-E (Fig. R1). In addition, we think it is reasonable to include $\langle -\nabla \cdot (\bar{\mathbf{u}}T') - \nabla \cdot (\mathbf{u}'\bar{T}) \rangle$ in the HFC-E as this term emerges in the governing equation of eddy temperature anomaly $\langle T' \rangle$ and vanishes in the non-eddy-resolving simulation.

Fig. R1 Contribution to the $\langle T_a \rangle$ change during the growing phase of the MHWs averaged at each grid point by $\langle -\nabla \cdot (\mathbf{u}'T') \rangle$ (a) and $\langle -\nabla \cdot (\bar{\mathbf{u}}T') - \nabla \cdot (\mathbf{u}'\bar{T}) \rangle$ (b). **c-d**, Same as **a-b**, but during the decay phase.

3. Why does MIX act to increase T_a ? I expect vertical mixing to be always cooling, since it represents a diffusive exchange with the colder ocean below 50 m.

As MHWs in this study are defined based on a seasonally varying threshold following Hobday et al. (2016), it is the anomaly of $\langle T \rangle$ relative to its climatological mean seasonal cycle (denoted as $\langle T_a \rangle$) that is related to MHWs. Accordingly, all the terms in the heat budget should be subtracted by their corresponding climatological mean seasonal cycles to quantify their induced changes of $\langle T_a \rangle$ during the MHW life cycles. When cooling induced by the vertical mixing is weaker than its climatological mean seasonal cycle, it contributes to the increase of $\langle T_a \rangle$.

4. Line 195: I do not understand the difference between Figs. 2 and 4. The analysis of large-scale MHWs is done by low pass filtering all terms in the Eq. (1), which should simply smooth the HFC-E. The smoothing cannot dramatically decrease the amplitude of HFC-E, unless the high-resolution field is extremely noisy and/or has alternating signs; none of these properties are visible in, for example, the Kuroshio region in Fig.2. It is possible that I am missing something, but the text needs to explain the difference.

Sorry for the unclarity. Fig. R2a shows a snapshot of the HFC-E in the Kuroshio extension on May 16th 1924. The instantaneous HFC-E is dominated by mesoscale variabilities with alternating signs, confirming that HFC-E represents the mesoscale eddies' effects. Therefore, spatially low-pass filtering does dramatically reduce the magnitude of HFC-E (Fig. R2b) and its contribution to MHW life cycles.

As evidenced by Fig. R2c and 2d, the positive (negative HFC-E) tends to occur in the growing (decaying) MHW phase. The altering signs of instantaneous HFC-E thus do not contradict with Fig. 2a and d which show the composited contribution of HFC-E to the temperature anomaly change during the growing and decaying phases of MHWs at individual grid points, respectively. The positive value in Fig. 2a (negative value in Fig. 2b) reflects that the HFC-E is generally positive (negative) during the MHW growing (decaying) phase.

We have provided Fig. R2a and b (Supplementary Fig. 9) in the revised manuscript. Please see Line 208-211.

Fig. R2 Snapshots of the (a) HFC-E and (b) spatially low-pass filtered HFC-E in the Kuroshio extension on May 16th 1924. (c) and (d) Same as (a) but only showing the HFC-E at the grid points where MHWs were growing and decaying on May 16th 1924, respectively.

Minor issues:

5. Line34: "... is comparable or larger than the scale of eddies"

Revised.

6. Line 37: comma before "as well as"

Revised.

7. Line 209 "the" is not needed

Revised.

8. Line 211: "in observations"

Revised.

9. Line 213-217: This sentence should be shortened or broken in two sentences.

Revised.

10. Line 294: it is better to say “climatological mean is subtracted from all the terms in Eq. (1)”

Revised.

11. Line 309: a prime is missing next to “u” in the formula

Corrected. Thanks.

REVIEWERS' COMMENTS

Reviewer #1 (Remarks to the Author):

To my view the issues raised during the first round of review have been satisfactorily addressed by the authors.

My recommendation is that the manuscript can be accepted.

Reviewer #2 (Remarks to the Author):

The authors adequately addressed all my original comments, and I recommend publication of this manuscript. A couple of minor issues need attention:

I.128: "evidently" is not needed here

I.230: "on average" should be placed before "longer"

Reply to the first reviewer

We are very grateful to you for your time in carefully reading our manuscript and providing helpful comments that make our manuscript better. We have carefully considered each of your comments (in blue) and revised the manuscript accordingly. Please find our response (in black) to your comments below.

Reviewer #1 (Remarks to the Author):

To my view the issues raised during the first round of review have been satisfactorily addressed by the authors.

My recommendation is that the manuscript can be accepted.

Thanks. We are grateful to your recommendation.

Reply to the second reviewer

We are very grateful to you for your time in carefully reading our manuscript and providing helpful comments that make our manuscript better. We have carefully considered each of your comments (in blue) and revised the manuscript accordingly. Please find our response (in black) to your comments below.

Reviewer #2 (Remarks to the Author):

The authors adequately addressed all my original comments, and I recommend publication of this manuscript. A couple of minor issues need attention:

Line.128: “evidently” is not needed here

Revised. Thanks.

Line.230: “on average” should be placed before “longer”

Revised. Thanks.